# The Recovery of Phosphate and Ammonium from Biogas Slurry as Value-Added Fertilizer by Biochar and Struvite Co-Precipitation

**Aftab Ali Kubar** [1,2,3,4,5], **Qing Huang** [1,2,3,4,5,*], **Muhammad Sajjad** [1,2,3,4,5], **Chen Yang** [1,2,3,4,5], **Faqin Lian** [1,2,3,4,5], **Junfeng Wang** [1,2,3,4,5,6] and **Kashif Ali Kubar** [7]

1  College of Ecology and Environment, Hainan University, Haikou 570228, China;
   kubar.aftabali@gmail.com (A.A.K.); sajjadsafi22@yahoo.com (M.S.); chen0307yang@163.com (C.Y.);
   lianfaqin1990@163.com (F.L.); drjunfengwang2010@163.com (J.W.)
2  Key Laboratory of Agro-Forestry Environmental Processes and Ecological Regulation of Hainan Province,
   Haikou 570228, China
3  Center for Eco-Environmental Restoration Engineering of Hainan Province, Haikou 570228, China
4  State Key Laboratory of Marine Resource Utilization in South China Sea, Hainan University,
   Haikou 570228, China
5  Key Laboratory for Environmental Toxicology of Haikou, Hainan University, Haikou 570228, China
6  College of Civil and Transportation Engineering, Shenzhen University, Shenzhen 518061, China
7  Department of Soil Science, Faculty of Agriculture, Lasbela University of Agriculture,
   Water and Marine Science, Uthal 90150, Baluchistan, Pakistan; kashifkubar@yahoo.com
*  Correspondence: huangqing@hainanu.edu.cn

**Abstract:** Biowaste materials could be considered a renewable source of fertilizer if methods for recovering P from waste can be developed. Over the last few decades, there has been a high level of interest in using biochar to remove contaminants from aqueous solutions. This study was conducted using a range of salts that are commonly found in biogas slurry ($ZnCl_2$, $FeCl_3$, $FeCl_2$, $CuCl_2$, $Na_2CO_3$, and $NaHCO_3$). Experiments with a biogas digester and aqueous solution were conducted at pH nine integration with $NH_4^+$, $Mg^{2+}$, and $PO_4^{3-}$ molar ratios of 1.0, 1.2, and 1.8, respectively. The chemical analysis was measured to find out the composition of the precipitate, and struvite was employed to remove the aqueous solutions. The study found that the most efficient removal of phosphate and ammonium occurred at pH nine in Tongan sludge urban biochar and rice biochar, respectively. Increasing the concentration of phosphate and ammonium increased the phosphate and ammonium content. Moreover, increasing the biochar temperature and increasing the concentration of phosphate and ammonium increased the efficiency of the removal of ammonium and phosphate. The removal efficiency of ammonium and phosphate increased from 15.0% to 71.0% and 18.0% to 99.0%, respectively, by increasing the dose of respective ions $K^+$, $Zn^{2+}$, $Fe^{3+}$, $Fe^{2+}$, $Cu^{2+}$, and $CO_3^2$. The elements were increased from 58.0 to 71.0 for $HCO_3^-$ with the increasing concentration from 30 mg $L^{-1}$ to 240 mg $L^{-1}$. This study concluded that phosphate and ammonium can be recovered from mushroom soil biochar and rice biochar, and phosphate can be effectively recovered via the struvite precipitation method.

**Keywords:** co-precipitation; removal efficiency; phosphates; ammonium; biochar; struvite

## 1. Introduction

Phosphorus and ammonium recovery are significant from an environmental management perspective as they contribute to eutrophication [1–6] and can be found in numerous wastewaters at variable concentrations [3,7–12]. Struvite ($MgNH_4PO_4.6H_2O$) is approximately 12% nitrogen (N) and 51.8% phosphorus (P). A $P_2O_5$ content greater than 30% is considered phosphate-rich. Therefore, a large amount of P recovery in struvite is particularly value added [4,13,14]. Additionally, struvite is also a comprehensive nutrient fertilizer

and, as such, can promote the germination and growth of vegetables [15–17]. At present, commonly used nitrogen and phosphorus fertilizers, such as diammonium phosphate (46% $P_2O_5$, 18% N) and triple superphosphate (48% $P_2O_5$), have similar nitrogen and phosphorus contents to struvite [13,18,19], which suggests that struvite could potentially be marketed as a fertilizer. The essential principle of the struvite precipitation method is to add a certain magnesium resource to the solution, so that the $NH_4^+$, $Mg^{2+}$, and $PO_4^{3-}$ in the solution are greater than the solubility product constant (Ksp). These three ions are converted to magnesium hexahydrate ammonium, and phosphate precipitation is recovered into N and P in solution [20,21]. Moreover, Cao et al. [14] suggested that struvite precipitation could remove ammonia, phosphate, and chemical oxygen demand (COD) substances from digested wastewater more efficiently and for a lower cost. Additionally, the recovery of heavy metals during rock phosphate processing as well as the extraction of other associated elements of value, such as ammonium and magnesium, is particularly significant considering resource efficiency and the sustainable use of biowaste materials [13–16]. Heavy metals recovered during phosphate fertilizer production are not dissipated onto agricultural soils from where they may leach into groundwater [17–21].

Furthermore, Tao et al. [22] studied the use of the struvite precipitation method to pretreat landfill leachate and reported that, when $n(Mg^{2+}):n(PO_4^{3-}):n(NH_4^+) = 0.8:1:1.4$, the ammonium removal efficiency can exceed 95%. When $n(Mg^{2+}):n(PO_4^{3-}) = 1.0$, most struvite crystals are produced, and there are a few non-crystalline ions. Common non-crystalline ions in biogas slurry, such as $Ca^{2+}$, $Zn^{2+}$, $Na^+$, $K^+$, and $CO_3^{2-}$, have different effects on the struvite precipitation reaction. Yuanyao et al. [23] reported that $Ca^{2+}$, $Na^+$, and $K^+$ in wastewater considerably inhibited the recovery of ammonia nitrogen, while $Ca^{2+}$ and $K^+$ significantly promoted the efficient total recovery of phosphorus [24]. Lujan-Facundo et al. [25] displayed their study results and noted that the maximum ammonium and phosphate removal (62.01% and 66.96%, respectively) were achieved with a molar concentration ratio ($Mg^{2+}:PO_4^{3-}$) of 2.8, pH of 10, and temperature of 22 °C. Another researcher found that the presence of $Ca^{2+}$ reduced the conversion efficiency of the struvite reaction, affected the purity of the struvite products, and inhibited the removal of $NH_3$–N. A high concentration of $Ca^{2+}$ in wastewater was found to have a very adverse effect on the recovery of nitrogen and phosphorus using struvite [26]. In addition, temperature also changed the activity of crystalline ions, thus, affecting the balance of chemical reactions [27]. When the reaction temperature of the struvite method rose from 10 °C to 50 °C, the Ksp of the struvite increased from $0.30 \times 10^{-14}$ to $3.73 \times 10^{-14}$ [28]. When the reaction temperature was lower than 20 °C, the reaction was slower and the removal efficiency of ammonia nitrogen was measured lower. The optimal temperature for the struvite precipitation method was found to be 25 °C [29]. When the temperature continued to increase to 45 °C, the removal efficiencies of ammonia nitrogen and phosphorus were almost unchanged [30–32]. For example, Huang et al. [33] suggested that the pH of the aqueous solution is a significant factor in the recovery of P using material precipitation. Besides this, they reported that the phosphate precipitation of $MgCl_2$ had a superior settling ability compared to $AlCl_3$ and $CaCl_2$ precipitation.

Several findings have been compiled to better remove phosphorus using the struvite precipitation method. Some researchers investigated the removal of $Mg^{2+}$ and $NH_4^+$ at a preliminary pH [8,34–37] and reported that approximately 97% and 85% of the $Mg^{2+}$ at different levels was removed at pH nine and eight, respectively, whereas struvite formation was found at pH seven. However, Adnan et al. [35] observed a higher efficiency of $PO_4^{3-}$ removal between pH 8.9 and 9.25 when testing anaerobic swine lagoon liquids in a variety of pHs and Mg:P ratios. Several scholars [38–43] reported that an increased Ca:P ratio in the struvite purity was less than 0.5 per pH 8.7. The lower productivity of the calcium phosphate and struvite combinations as fertilizer appears to be due to the scarcity of ammonium ions in the mixture compared to pure struvite and the development of small precipitates.

The focus on biogas slurry when it is used as a fertilizer focuses on the effects of its application, including improving crop productivity, product quality, crop resistance, and soil fertility [44–50]. In order to better understand the use of biogas slurry, it is important to fully recognize and explain biogas slurry's characteristics and value [51]. Determining a worth for biogas slurry is an essential aspect of this study in order to establish and promote the value of biogas slurry as a fertilizer [52–55]. Although researchers understand the value of biogas slurry and have started to consider the problems of reprocessing and better utilizing biomass resources in agriculture, presently, few studies have assessed biogas slurry at the domestic and international levels [3,6,11,56].

Biochar has a great capacity for adsorbing cations and anions from a solution containing a variety of organic compounds. Using biochar to recover surplus nutrients from biogas slurry effluents and then utilizing the nutrient-enriched biochar for soil alteration offers an eco-friendly solution to multiple issues [57–59]. Biochar has a certain adsorption capacity for ammonia nitrate. The adsorption capacity of ammonia nitrate is about 1 mg g$^{-1}$ [60–63]. The accessibility of the adsorption positions is ruled by the adsorbent dose as suggested by Rafeah et al. [64].

The main factors influencing the formation of struvite crystals include temperature, pH, reaction time, crystalline ions, and non-crystalline ions. The activity of conformational ions at different pH levels differs, which changes the supersaturation, which, in turn, affects the driving force of struvite formation [27]. However, a higher quantity of struvite should also be produced where $Mg^{2+}$ is the only limiting ion. In the appropriate pH range, the greatest amount of struvite is produced [65–67]. Therefore, in the process of biogas slurry treatment, the pH range should be controlled at all times. Dongmei et al. [30] used the struvite method to remove ammonia nitrogen from pig farm wastewater. Laridi et al. [68] reported that, when the pH was ten, the removal efficiency of ammonia nitrogen reached 87%, and the residual phosphate concentration was correspondingly the lowest at 30.21 mg L$^{-1}$. In another study, the optimum pH range for the struvite method was measured to be between eight and ten, and the removal efficiencies of phosphate and ammonia nitrogen reached 98% and 17%, respectively [69]. When the pH reached 7–11, a large number of magnesium ammonium phosphate byproducts were formed. The utilization response method augmented the process conditions of the struvite method [70].

Recently, sustainability researchers have been exploring using struvite precipitation in various types of surplus water, such as pig sewage waste [71,72], anaerobic effluent [35,73,74], landfill leachate waste [75,76], and synthetic wastewater as a method of $NH_4^+$. However, the potential for $NH_4^+$ and $PO_4^{3-}$ removal and $PO_4^{3-}$ recovery from different biogas digester waste using struvite precipitation has not been meritoriously examined. The pH value controls the accessibility of soluble$NH_4^+$, $PO_4^{3-}$, and $Mg^{2+}$ concentrations [15]. $Mg^{2+}$ is a restrictive ion in the digester effluent waste, which reviews the impact of the pH. The$Mg^{2+}$:$PO_4^{3-}$ molar ratio modification is key to the optimum recovery of nutrients. A major benefit of removing phosphorous from wastewater via the struvite precipitation method is that the phosphorus and ammonia are readily soluble and accessible to plants. The recovered phosphorus can consequently be used as fertilizer [77,78].

Therefore, the main objectives of this study were to assess the recovery of phosphate and ammonium nitrate from biogas slurry effluent via struvite co-precipitation and determine the influence of pH and $Mg^{2+}$ and $PO_4^{3-}$ molar ratio modification on the struvite co-precipitation. Additionally, the study examined the effects of different variables, such as pH, $PO_4^{3-}$, $NO_3^-$, $Fe^{2+}$, $Fe^{3+}$, $CO_3^{2-}$, $HCO_3^-$, $Zn^{2+}$, $K^+$, and $Cu^{2+}$, on the phosphorous and ammonium recovery capacity of biochar and their availability to plants.

## 2. Materials and Methods

### 2.1. Molar Ratio and pH Adjustment

The artificial wastewater for the study was collected from Hainan University marine resource utilization laboratory in the South China Sea, Haikou, China. The water quality

parameters of the aqueous solution were analyzed. The $NH_4^+$: $Mg^{2+}$:$PO_4^{3-}$ molar ratio of the effluent was maintained. The experiments at ions were accompanied to evaluate the struvite precipitation prospective of the tested aqueous solution. The experiment was carried out with $NH_4^+$: $Mg^{2+}$:$PO_4^{3-}$ molar ratio amendment, but the pH was adjusted by 1 mol $L^{-1}$ NaOH to produce a pH range of 8.4–9.6. The maximum pH for the entirety of the experiment was established at nine to avoid $NH_3$ emissions [79,80]. Moreover, the pH for lesser solubility of struvite was expected for most of the waste waters were nine [35,81]. The experiment was conducted using $NH_4^+$: $Mg^{2+}$:$PO_4^{3-}$ in a ratio of 1:1.2:1.8, at pH nine. The solutions were rotated at 200 rpm and agitated for 20 min. The pH was also examined during the course of the experiments. To investigate the changes in concentration of $Mg^{2+}$, $NH_4^+$, and $PO_4^{3-}$ ions during the precipitation process, the samples were filtered by a 0.22 µm filter to avoid from receiving any minute struvite crystals that may have been created during sampling.

### 2.2. Details of the Experiments

To prepare the $K_2HPO_4$ stock solution, 28.5275 g of $K_2HPO_4.3H_2O$ was weighed precisely, dissolved in an appropriate amount of water, and then water was added until the volume was 250 mL. The solution concentration was 0.5 mol $L^{-1}$. The phosphate solutions needed for the experiments were obtained by diluting the stock solution. The pH standards for the adsorption solutions were maintained at pH six by using 0.1 mol $L^{-1}$ NaOH and 0.1 mol $L^{-1}$ HCL. The phosphorus adsorption reactions were conducted in batches to enhance the factors, including an initial pH of nine, a content time of 20 min, and initial concentrations of 30, 60, 100, 160, and 240 mg $L^{-1}$ for the adsorbents (0.2 g), and the phosphate solutions were poured into 50 mL conical flasks. The mixtures were shaken for 20 min at 200 rpm under a constant temperature of 25 °C. The remaining phosphate in the filtrate with syringe filters was 0.45 µm. The retained phosphate in the filtrate was calculated via the ammonium molybedate spectrophotometric method using a UV-650 nm spectrophotometer.

To prepare the $NH_4Cl$ stock solution, 6.6863 g $NH_4Cl$ was precisely weighed, dissolved in an appropriate amount of water, and then water was added until the volume was 250 mL. The solution concentration was 0.5 mol $L^{-1}$. The initial concentration of ammonium nitrate was 252 mg $L^{-1}$. A total of 0.2 g of biochar was combined with the solution and mechanically stirred at 200 rpm at 25 °C. The pH of the solutions was adjusted to 0.1 mol $L^{-1}$ NaOH and six using 0.1 mol $L^{-1}$ HCL. The $NH_4$–N concentrations were measured with an ultraviolet spectrometer in order to determine the $NH_4$–N content. The Nessler technique was also used to measure the samples' $NH_4$–N concentrations. The wavelength useful to spectroscopy was UV-450 nm.

The activated carbon employed in this study was obtained from Hainan Province (Haikou) at different pyrolysis temperatures. The particle size of this activated carbon ranged from 0–4 mm. All the experiments were carried out in Hainan University's main marine resource utilization laboratory in China. Generally, the following instruments were used to find the porosity and texture of the biochar used in the experiment. A spectrophotometer and pH meter were used in our experiments in order to find the ammonium nitrate and phosphate removal efficiencies. The details of the biochar used in the experiments are shown in Table 1.

### 2.3. Chemical Analysis

The chemical solution was clarified after ten minutes of the experiments' duration by a 0.2-µm filter. Phosphate ($PO_4^{3-}$) was determined by the ascorbic acid ammonium molybdate method, and ammonium nitrate was determined by the Nessler technique using a spectrometer. All the experiments were replicated three times and the means were reported.

### 2.4. Phosphate and Ammonium Removal Efficiencies

The phosphate adsorption capability of the adsorbent and the removal efficiency were calculated using the following equation.

$$Removal\ (\%) = \frac{(Ci - Ce)}{Ci} \times 100 \tag{1}$$

*Ci* and *Ce* represent the initial and final concentrations of phosphate.

The removal efficiency of ammonium nitrate (%) was usually determined using the Equation (1). *Ci* and *Ce* are the initial and final concentrations (mg L$^{-1}$), respectively.

**Table 1.** Biochar used in the experiments.

| Sample Number | Biochar Type |
| --- | --- |
| CK | Blank |
| DG | Bean stem biochar |
| SH-1 | Water hyacinth root biochar |
| SH-2 | Water hyacinth stem and leaf biochar |
| HSK | Peanut shell biochar |
| MG-1 | Mushroom soil biochar (first batch) |
| MG-2 | Mushroom soil biochar (second batch) |
| SD | Rice Biochar |
| JM | Jimei block sludge biochar |
| TA-1 | Tongan City Sludge Biochar |
| TA-2 | Tongan Hydrothermal Sludge Biochar |
| TA-3 | Tong'an Hydrogel Sludge Biochar |
| XJ | Rubber wood biochar |

## 3. Results

### 3.1. Recovery Efficiencies of Phosphate and Ammonium Using Biochar Contents

The recovery efficiencies of P and NH$_4^+$ using 12 different biochars were tested. The biochars were produced at different pyrolysis temperatures. The results showed that the removal efficiency of NH$_4^+$ was highest (33%) in TA-1 (Tongan city sludge biochar), followed by SD (rice biochar) and TA-3 (Tongan hydrogel sludge biochar), and was lowest (12.5%) in JM (Jimei block sludge biochar). This was followed by SH-1(water hyacinth root biochar) (Figure 1a). However, the removal efficiency of P was most efficient (45.36%) using SD (rice biochar), which was followed by MG-2 (mushroom soil biochar) and JM (Jimei block sludge biochar). The lowest P removal efficiency (30.26%) was recorded in SH-1 (water hyacinth root biochar), which was followed by DG (bean stem biochar) (Figure 1b). The struvite crystals had an adsorption effect, thereby, promoting the struvite reaction to proceed. Therefore, the removal efficiency of ammonia nitrogen slightly increased. However, the water hyacinth root biochar, water hyacinth stem, and leaf biochar as well as Jimei block sludge biochar had no clear effect on the ammonia nitrogen recovery efficiency of struvite precipitation. This could be because the adsorption ability of ammonia nitrogen and the struvite crystals was weak.

### 3.2. Recovery Efficiency of Phosphate and Ammonium at Different Temperature Intervals

The removal efficiency of NH$_4^+$ increased with higher temperatures. The removal efficiencies of NH$_4^+$ were 15.03%, 15.92%, 22.61%, 30.03%, 28.44%, and 30.19% at control CK, MY, T300, T500, T700, and T900, respectively. The residual removal efficiencies of ammonium were 32.11%, 44.97%, 29.25%, 26.60%, 27.32%, and 26.44% at control CK, MY, T300, T500, T700, and T900, respectively (Figure 2a). Similarly, the removal efficiency of P increased at 700 °C and decreased at 900 °C. With a rise in temperature, the P removal efficiencies were 30.09%, 32.65, 31.51%, 46.01%, and 46.79% before decreasing to 44.87% under control CK, MY, T300, T500, T700, and T900, respectively. The residual removal efficiencies of P were 32.50%, 51.67%, 38.02%, 39.30%, 44.11%, and 46.14% at control CK, MY, T300, T500, T700, and T900 (Figure 2b). After adding it to the simulated biogas slurry,

the initial phosphorus and ammonia nitrogen concentrations in the biogas slurry increased, while the magnesium ion concentration was not affected.

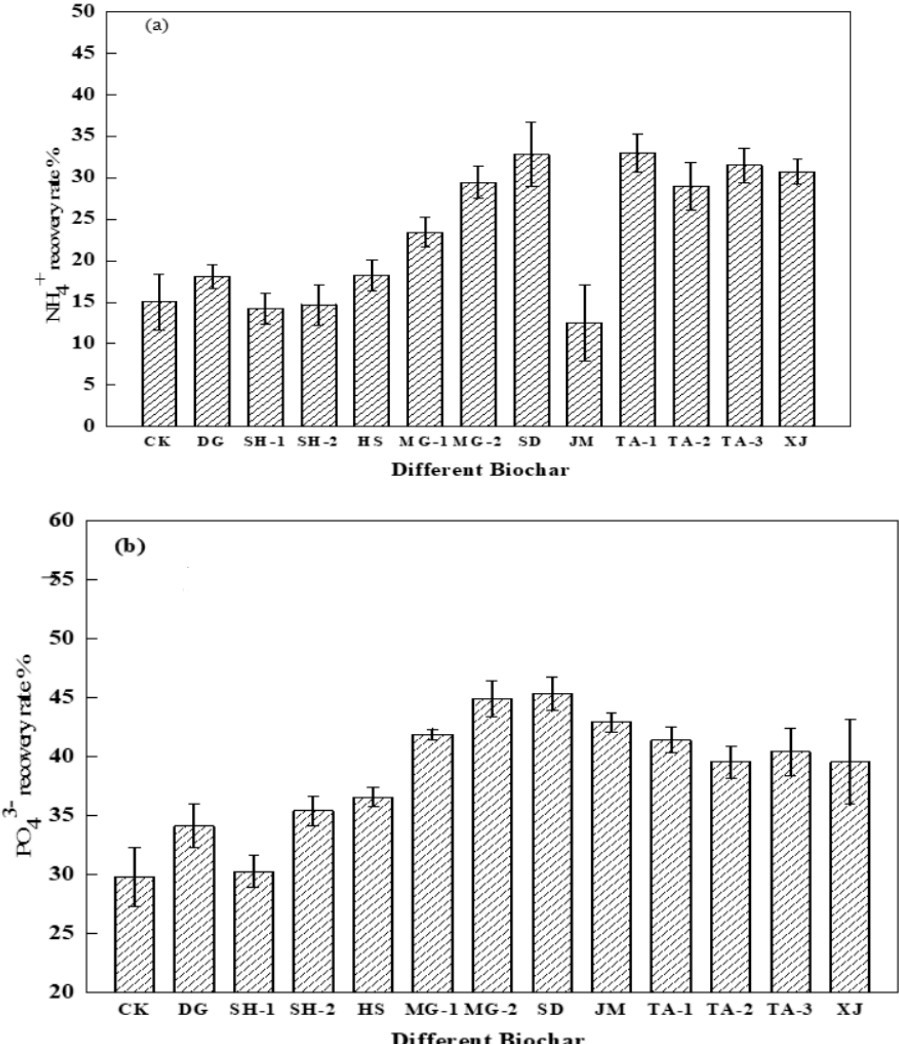

**Figure 1.** Effects of biochar types on ammonium and phosphorus recovery: (**a**) ammonium recovery efficiency and (**b**) phosphorus recovery efficiency.

*3.3. Effects of Initial Phosphorus Concentration on Phosphate Recovery, Ammonium Recovery, and Their Residual Concentrations Using the Struvite Precipitation Method*

The removal efficiencies of phosphate and ammonium and the residual concentration of P and $NH_4^+$ were examined in this study. The highest P removal efficiency (86.5%) was observed with a P concentration of 124 mg $L^{-1}$, whereas the lowest removal efficiency (18.88%) was recorded with a P concentration of 31 mg $L^{-1}$. The highest residual removal efficiency of phosphate was noted to be 20.65% at 124 mg $L^{-1}$, and the lowest residual removal efficiency of phosphate was 8.21% at 1.55 mg $L^{-1}$ (Figure 3a). The results of the $NH_4^+$ removal efficiency indicated that the highest $NH_4^+$ removal efficiency (45.61%) occurred at an $NH_4^+$ concentration of 100 mg $L^{-1}$. However, the lowest $NH_4^+$ removal efficiency was 15% at an $NH_4^+$ concentration of 5 mg $L^{-1}$. The highest residual removal efficiency for ammonium was 59.73% at 124 mg $L^{-1}$, and the lowest residual removal efficiency was 1.59% at 1.55 mg $L^{-1}$ (Figure 3b). The P and $NH_4^+$ recovery efficiencies increased with arise in the initial P concentration. Concentrated biogas slurry is more favorable to the recovery of phosphorus and nitrogen, which is mainly due to ion activity.

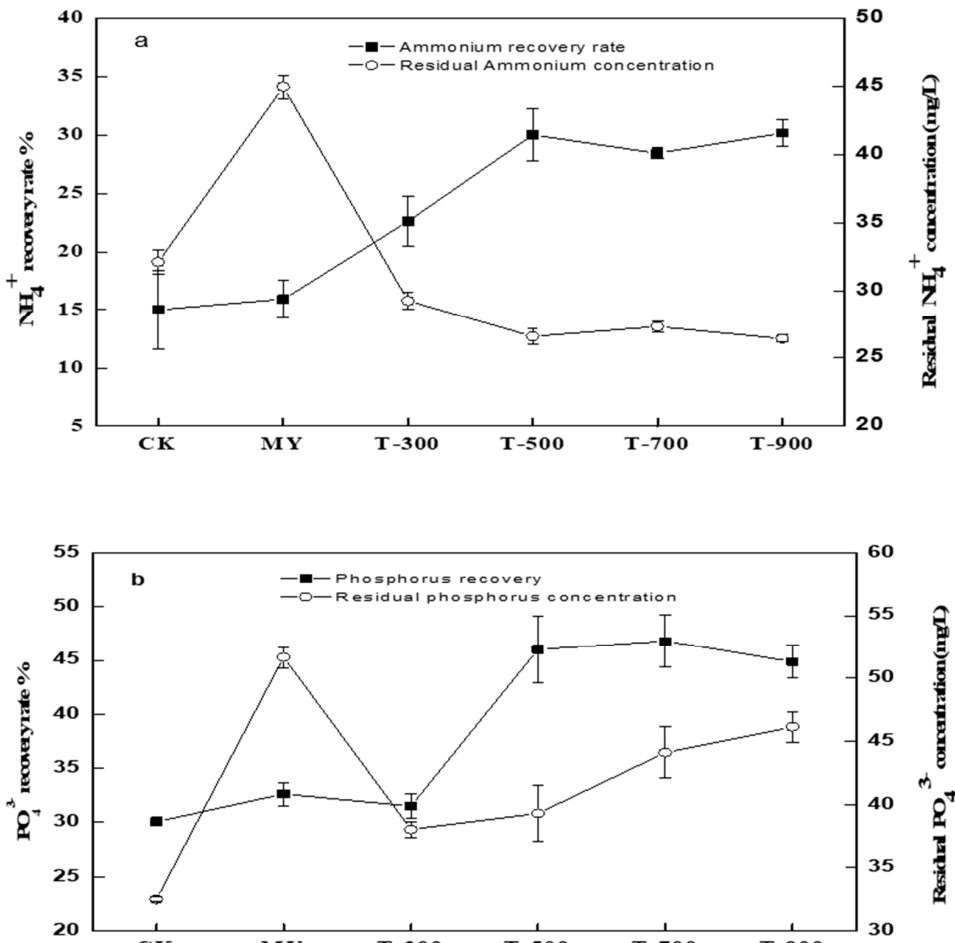

**Figure 2.** Effect of temperature on phosphate and ammonium recovery: (**a**) ammonium recovery efficiency and (**b**) phosphorus recovery efficiency.

*3.4. Effects of Different Ion Concentrations on the Removal Efficiencies of Phosphate and Ammonium*

The removal efficiency of ions was reduced by increasing the initial concentration of ammonium ions. The removal efficiencies were observed to decrease in the ranges of 40.12–23.81%, 55.32–46.23%, 45.75–13.13%, 43.18–16.86%, 51.55–40.14%, 40.07–22.85%, and 66.41–54.36% for $K^+$, $Cu^{2+}$, $Zn^{2+}$, $Fe^{3+}$, $Fe^{2+}$, and $CO_3^{2-}$, respectively. The highest residual removal efficiency of $NH_4^+$ was 2.54%, and the lowest was 1.27% using a concentration of $K^+$ ions (Figure 4a–f). The elements increased to 57.99–71.07% for $HCO_3^-$ when the concentration increased from 30 mg $L^{-1}$ to 240 mg $L^{-1}$ (Figure 4g). Furthermore, the removal efficiency of phosphate also decreased when the initial concentration of metal ions was increased. The removal efficiencies dropped from 99.17% to 98.39%, 97.36% to 96.71%, 97.27% to 96.56%, 97.59% to 95.91%, 97.56% to 97.40%, and 98.32% to 97.41% for $K^+$, $Fe^{3+}$, $Fe^{2+}$, $CO_3^{2-}$, $HCO_3^-$, and $Cu^{2+}$. The highest residual removal efficiencies of phosphate observed were 2.71% and 5.30% in a concentration of $K^+$ ions (Figure 5a–f). The phosphate removal efficiency of the metal ions increased from 98.76% to 99.26% for $Zn^{2+}$ when the concentration was increased from 30 mg $L^{-1}$ to 240 mg $L^{-1}$ (Figure 5g).

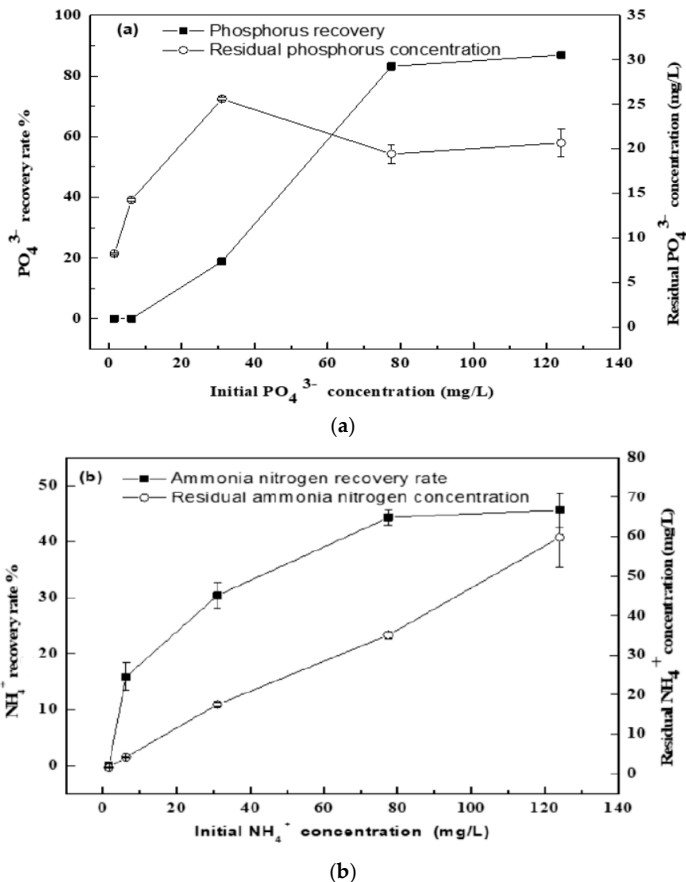

**Figure 3.** Effects of initial phosphorus concentration on phosphate recovery, ammonium recovery, and their residual concentrations using the struvite precipitation method: (**a**) phosphorus recovery efficiency and (**b**) ammonium recovery efficiency.

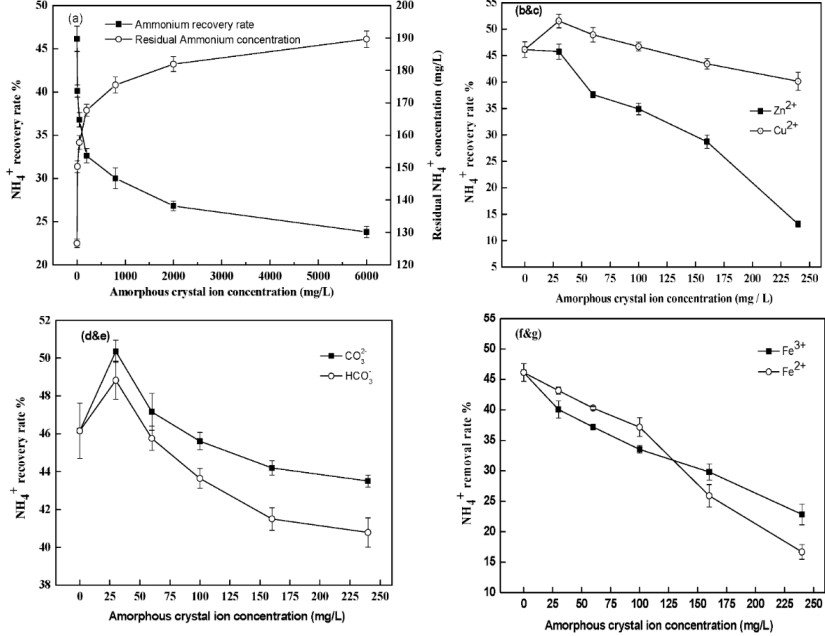

**Figure 4.** Effects of ions on ammonium recovery: (**a**) $K^+$, (**b,c**) $Zn^{2+}$ and $Cu^{2+}$, (**d,e**) $CO_3^{2-}$ and $HCO_3^-$, (**f,g**) $Fe^{3+}$ and $Fe^{2+}$.

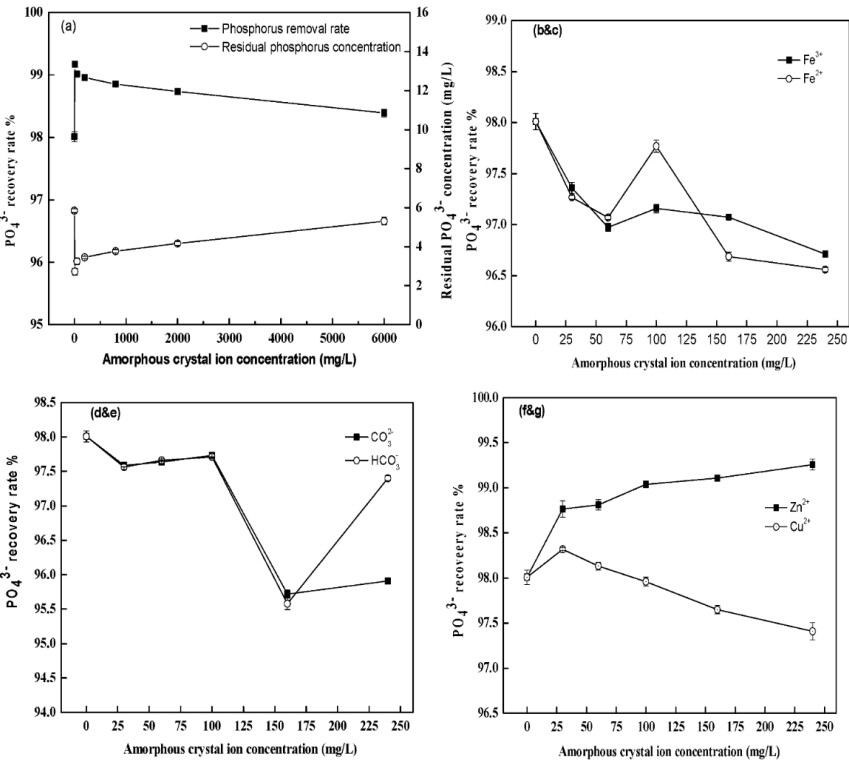

**Figure 5.** Effects of ions on phosphate recovery: (**a**) $K^+$, (**b,c**) $Fe^{3+}$ and $Fe^{2+}$, (**d,e**) $CO_3^{2-}$ and $HCO_3^{-}$, and (**f,g**) $Zn^{2+}$ and $Cu^{2+}$.

## 4. Discussion

### 4.1. Recovery Efficiency of Phosphate and Ammonium by Biochar Contents

In this study, the P and $NH_4^+$ recovery efficiencies using 12 different biochars were tested. This study revealed that the $NH_4^+$ removal efficiency was highest using TA-1 (Tongan city sludge biochar) and lowest using JM (Jimei block sludge biochar). Meanwhile, the highest P removal efficiency was noted using SD (rice biochar) and the lowest using SH-1 (water hyacinth root biochar). These findings echo those reported by Elsevier researchers. They observed that the phosphate removal efficiency gradually increased when the content of the biochar increased [82–85]. However, this study assessed the recovery of $NH_4^+$ and P from biogas slurry in combination with biochar using struvite precipitation. The P and $NH_4^+$ recovery efficiencies reached 33% and 45%, respectively, and were stable at an optimal pH of nine using 12 different types of biochar. $NH_4^+$ and P recovery efficiencies were better using struvite precipitation and the biochar adsorption mechanism. These results demonstrate that biochar is a good source for struvite precipitation to recover P and $NH_4^+$ from biogas slurry as a slow-release fertilizer. Similar to this study's results, Zheng et al. [85] found that the recovery of P and N from urine increased by approximately 40–50% and 99%, respectively, with the combination of biochar and struvite at pH nine. The communication time between the biochar and nutrients solution had a considerable impact on the reactivity of the biochar and element sorption [86]. Generally, biochars created by slow pyrolysis show greater S, $Na^+$, P, $Ca^{2+}$, $Mg^{2+}$, cation exchange capacity (CEC), and surface area than biochars created via fast pyrolysis [87]. Increasing the temperature generally results in reducing the production of biochar but enhancing the biochar's total $K^+$, $Ca^{2+}$, and $Mg^{2+}$ content, as well as pH, ash content, and surface area. It also reduces the biochar's cation exchange capacity [56,87,88] suggesting that activated and nutrient-enriched biochars could be valuable for absorbing the soil's N and P, thus, reducing losses through leaching.

### 4.2. Recovery Efficiency of Phosphate and Ammonium at Different Temperature Intervals

In this study, higher phosphate and ammonium removal efficiencies were noted at T900 temperatures. The removal efficiency was generally improved by increasing the temperature. This finding is consistent with other researchers [3,89,90]. A previous study [91] recognized that biochar produced at low combustion temperatures, such as 300 °C, may restrain N from solution because of the support of higher microbial movement affected by greater amounts of bioavailable P in its structure. The reason may be that, as the pyrolysis temperature increases, the phosphate and ammonia nitrogen content of the biochar itself increases accordingly, and the nitrogen and phosphorus content adsorbed during the reaction also increases accordingly [92,93]. The struvite particles are stable in a structure at temperatures below 55 °C at which point the quantity loss arises. This is because the solubility of struvite particles increases with a rising temperature. The rise in temperature during the pyrolysis process results in a rise in the carbon content and a reduction in the oxygen and hydrogen content. This is due to the dissipation of functional compounds comprising hydrogen or oxygen. Most of the earlier studies reported that the adsorption of pollutants by adsorbents seemed to be an endothermic process and the adsorption ability increased in line with higher pyrolysis temperatures [94–96]. However, Huang et al. [97] revealed that the maximum removal efficiency was noted when pyrolysis was repeated five times at pH 8.0–8.5. While Reference [98] observed that the ammonium removal efficiency (mass) decreased from 92% to 77% after five recycles. Struvite could also be efficiently decomposed using sodium hypochlorite. The mechanism planned was combined dissolution and oxidation of struvite [33]. Masuda et al. [99] found that biochar retained 78% to 91% of the absorbed $NH_4^+$ and 60% of the absorbed $PO_4^{3-}$ at response times under 24 h. The pyrolysis temperature has a significant effect on the physicochemical properties of various biochar formed from feedstock materials. Biochar from plant waste had an essential effect on the removal efficiencies of phosphate and ammonium. The lower negative load and higher surface area of the biochar created a higher adsorption capacity for $NO_3^-$ and P, which ultimately increased the removal efficiency [58]. This study's results show that the use of TA-1 and SD can be an effective way of eliminating P and $NH_4^+$ from biogas slurry through adsorption.

### 4.3. Effects of the Initial Phosphorus Concentration on Phosphate Recovery, Ammonium Recovery, and Their Residual Concentrations through the Struvite Precipitation Method

In this study, the highest P removal efficiency (86.5%) was observed using a P concentration of 124 mg $L^{-1}$, whereas the lowest (18.88%) was recorded using a P concentration of 31 mg $L^{-1}$. The highest removal efficiency of $NH_4^+$ (45.61%) was noted using an $NH_4^+$ concentration of 100 mg $L^{-1}$. However, the lowest removal efficiency of $NH_4^+$ (15%) occurred using an $NH_4^+$ concentration of 5 mg $L^{-1}$. The P and $NH_4^+$ recovery efficiencies increased when the initial P concentration increased. This is possibly due to the higher concentration of biogas slurry, which encourages the recovery of nitrogen and phosphorus due to ion activity. According to the supersaturation calculation formula, the supersaturation improved with the increasing rate of ion activity, and supersaturation will affect the nucleation induction time and crystal growth rate of the struvite [33,96,100–106]. According to the consequences of the same ion, enhancing the concentration of nutrient ions could promote the forward reaction of struvite precipitation, thereby, improving the recovery efficiency of NH4+ and P [96,104]. However, Wang et al. [107] identified that, when the concentration of phosphate is reduced from 100 to 20 mg $L^{-1}$, the precipitation rate declined from 396.65 mg $L^{-1}$ to 70.46 mg $L^{-1}$, demonstrating that the struvite reaction rate could be managed by regulating the pH according to the modification of the P concentration. It is well documented that struvite precipitation is the preferred biochemical technique for $NH_4^+$ and P recovery from different aqueous solutions, with important potential significance for nitrogen recovery [14,108–110]. This is a highly effective and biologically approachable chemical method for recovering nitrogen for use as fertilizer [16]. Escudero et al. [76] reported struvite precipitation at a 1:1:1 ratio removed

95% of $NH_4^+$ from anaerobic effluents in 30 s. Meanwhile, several other researchers have confirmed that struvite is favorable for the recovery of phosphorus and ammonium from livestock manure, urine, industrial waste water, anaerobically treated effluents, and landfill leachates [9,67,97,111–113]. Therefore, recovering nitrogen and phosphorus from biogas slurry through struvite represents a desirable approach. Struvite precipitation is likely an effective approach for recovering ammonium and phosphate content from biogas slurry and waste water, achieved in the form of solid complex precipitation. This precipitant is a valuable multi-nutrient, slow-release fertilizer for the production of vegetables and crops.

### 4.4. Effects of Different Ion Concentrations on the Removal Efficiencies of Phosphate and Ammonium

The effectiveness of removing $NH_4^+$ ions reduced when the initial concentration of ions was increased. These results are in agreement with other investigators who reported that removal efficiencies of metal ions increased with rising concentrations [19,23,84,97,113–115]. This decline in the removal efficiency of metals is due to the lower number of adsorbent surface sites and, consequently, the metals cooperating with the adsorbent sites could be easily removed [116]. The removal efficiencies of phosphate and $NH_4^+$ are not directly proportional to the biochar specimen or to the biochar dose. This is an overall fact about the adsorption method, which is mostly due to the overlap of adsorption sites based on the increase in solids [117–119]. The concentrations of $K^+$ and $Na^+$ during storage were 842.68 mg $L^{-1}$ and 2151.12 mg $L^{-1}$, respectively, which were very high relative to the recorded values [24,78,120]. However, the compositions of both $Na^+$ struvite and $K^+$ struvite were not impacted by the difference in pH, but the probability of $K^+$ struvite produced relative to $Na^+$ struvite improved by increasing the pH. The optimum pH for $Na^+$ struvite was 12, and the $K^+$ struvite was very soluble and precipitated readily with an increase in pH. The molar ratio for $Na^+$ and $K^+$ was greater than ten, which was favorable to obtain the precipitant and was very soluble as the pH increased [121–123].

In the meantime, the recovery efficiencies for ammonia stayed fairly stable, whether with or without the additional consumption of Mg salt. This could be clarified by owing to the unnecessary inclusion of salt [22,123–126]. In addition, the influence of an organic source, such as humic acid, on the absorption potential of phosphate has been thoroughly examined in the literature [127,128]. Regarding diverse anions, anions with bivalency had a higher load density of, for example, $SO_4^{2-}$ and $CO_3^{2-}$, which have a higher influence on the adsorption potential of phosphate than monovalent anions, such as $Cl^-$ and $NO_3^-$. The occurrence of $CO_3^{2-}$ ions has been shown to greatly decrease the adsorption potential of phosphate in many experiments [59,94,129]. Moreover, Jing et al. [130] reported that there was a noteworthy reduction in the adsorption ability of phosphate onto $Mg^{2+}$ modified biochar when it cohabits with $CO_3^{2-}$ ions, which actually qualified for competition with $Mg^{2+}$ ions to form amorphous magnesium carbonate. Comparable explanations were also reported for the decrease in the removal of phosphate in the existence of $HCO_3^-$ ions [131–133]. The presence of viable ions in solutions, such as $Ca^{2+}$, $K^+$, $Na^+$, $CO_3^{2-}$, and $HCO_3^-$, will powerfully impact the nucleation and development of struvite [106,134–136].

### 4.5. Commercialization and Applicability of This Study as Value-Added Fertilizer

Compared to commercial mineral P fertilizers, struvite is an effective slow-release fertilizer with a comparatively low content of contaminants, which could be the best alternative as the fertilizers are produced from phosphate rock [15,18,19,21]. The worthiness of struvite as a fertilizer has only recently been understood and it is now the focus of increasing research attention [106,123]. A perfect technology would feature maximum P recovery rates, good removal, and an applicable material with low environmental risks, good fertilizing effects, and financial efficiency. This work proves that P and $NH^{4+}$ recovery can be achieved with low costs. In some cases, even financial gains from P and $NH^{4+}$ recovery can be achieved if dissolved P is recovered from a biogas aqueous supernatant. Small quantities of recovered struvite are currently being tested on ornamental plants as fertilizer. It is essential to assess the efficiency of struvite on different types of crops

and plant growth as a substitute source of fertilizer. The fertilizer properties of struvite have been examined by several researchers [70,74,113]. From the last decade, struvite had been commercially produced in Japan, and sold to fertilizer companies [22]. It is very effective for those crops, which required low-soluble fertilizers. This work exposes that recovery of P can be cost-neutral under certain conditions if dissolved P is recovered as struvite or calcium phosphate from the aqueous phase. Field trails validate that struvite and different forms of calcium phosphate, for example, have a relative fertilizer efficacy in different soils equivalent to a water-soluble commercial single super phosphate [111]. However, diverse studies expose that the plant availability is not solely influenced by the quality of the recovered product. In fact, natural soil properties, such as pH, P supply, and type of vegetation, significantly influence the plant uptake [44,45]. Consequently, further field trials are also prerequisite to observe their actual fertilizing effect and especially their long-term behavior in the soil.

## 5. Conclusions

This study examined phosphate and ammonium recovery efficiencies using biochar and the struvite precipitation method. Different ion concentrations were used as the precipitators for the recovery of P and $NH_4^+$ from biogas slurry. The P and $NH_4^+$ removal efficiencies were significantly influenced by the pH, temperature, biochar type, biochar content, and different concentrations of elements, such as $K^+$, $Zn^{2+}$, $Fe^{3+}$, $Fe^{2+}$, $Cu^{2+}$, $CO_3^{2-}$, and $HCO_3^-$. The most P was recovered using $ZnCl_2$, more so than when any other metal salt was used. More $NH_4^+$ was recovered using $NaHCO_3^-$ compared to other metal salts. The maximum P and $NH_4^+$ recovery efficiencies were 45.36% (using SD) and 33% (using TA-1), respectively, at the adsorbent dosage of 0.2 g. The removal efficiency of P and $NH_4^+$ increased with temperature and different concentrations of metal salts. Furthermore, the findings from this study suggest that struvite could be a favorable adsorbent for effectively and efficiently removing phosphate and ammonium from aqueous solutions. This study clarified the mechanisms by which biochar efficiently recycles nitrogen and phosphorus from biogas slurry and then provides a sustainable solution for agriculture. Developing the previously mentioned biochar and using it as fertilizer would reduce the economic cost of fertilizers in China. This precipitation technique could also help solve the problems of disposing urban and rural biogas slurry and poultry waste materials.

**Author Contributions:** Conceptualization, Q.H. & A.A.K. Methodology, A.A.K. Software, C.Y. Validation, J.W. Formal analysis, C.Y. and K.A.K. Investigation, Q.H. Resources, Q.H. Data curation, C.Y. and K.A.K. Writing—original draft preparation, A.A.K. Writing—review and editing, M.S. Visualization, F.L. Supervision, Q.H. Projected ministration, Q.H. Funding acquisition, Q.H. All authors have read and agreed to the published version of the manuscript.

**Funding:** This study were funded by the National Nature Science Foundation of China (41571288), Hainan Pro-vincial Natural Science Foundation of China (319MS008), National Science and Technology Support Program of China (2014BAD14B04), National Natural Science Foundation of China (41867047), and Research Initiation Fund of Hainan University (KYQD(ZR)20032).

**Institutional Review Board Statement:** Not applicable.

**Informed Consent Statement:** Not applicable.

**Data Availability Statement:** Not applicable.

**Acknowledgments:** We gratefully thank our team members of Qingqing Wang, Genmao Guo, and Yin Liu, Who read and modified the manuscript, and the Rothamsted Research David Stephen Powlson for his critical reading of the manuscript.

**Conflicts of Interest:** The authors declare no conflict of interest.

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
