# Peer review of "The Recovery of Phosphate and Ammonium from Biogas Slurry as Value-Added Fertilizer by Biochar and Struvite Co-Precipitation"

_sustainability, doi:10.3390/su13073827_

Round 1
Reviewer 1 Report
Thank you for submitting this work to sustainability. I think this is a good fit and I hope we can publish this soon.
Below a few comments I would like you to consider as I think that they can further improve the quality of your work.
General:
-check for typos and minor language mistakes. I found quite a few and you do not want this to affect your final paper
Abstract:
-I strongly disagree with the idea that P will be depleted in the upcoming next century - this is highly contested among scientists - having an alternative to mineral phosphates in form of Biochar is strong point - I would not dive into the P scarcity debate here if I were you
Introduction:
-You make a strong point for struvite and alternatives to mineral fertilizers. There is a recent study on heavy metals in mineral fertilizers (https://doi.org/10.1016/j.rser.2021.110740). Not using mineral phosphates may decrease heavy metal dissipation through fertilizers. This can also be added in the introduction to make your point even stronger
Materials and Methods:
-Where exactly came the activated carbon from. A number of biowaste materials are presently considered to produce active carbon, so this could be important to some readers
-ok you have that in table 1 - good
Results:
-line 235 check typo
-Fig 1 - I would call this extraction rate since you are not recovering it yet - it is with the biochar
-provide all figures in higher quality
Discussion:
-include an assessment of the industrial applicability - these results are very interesting, now if I want to apply this on industrial scale, how would it look like?
Author Response
Reviewer # 1
Thank you for your valuable comments. We have revised this manuscript according to your suggestions. I hope that the revised manuscript would meet your requirements for Publication in Sustainability. Based on your suggestion and comments, we have carefully modified the manuscript (highlighted with green color for reviewer two and track changes for professional English editing). We wish that the revised manuscript will meet the standard for publication.
Comment: Thank you for submitting this work to sustainability. I think this is a good fit and I hope we can publish this soon. Below a few comments I would like you to consider as I think that they can further improve the quality of your work.
Response: Thank you for your comments and valuable suggestions. We have carefully revised the manuscript according to your suggestions and comments
Comment: General: -check for typos and minor language mistakes. I found quite a few and you do not want this to affect your final paper
Response: Thank you for valuable comments. We have carefully rechecked the whole manuscript to avoid unnecessary errors in the revised manuscript. The writing of manuscript has been improved, the language flow of the manuscript has been checked by the native speaker in this document, and most of the errors and inconsistencies has been improved.
Comment: Abstract: -I strongly disagree with the idea that P will be depleted in the upcoming next century - this is highly contested among scientists - having an alternative to mineral phosphates in form of Biochar is strong point - I would not dive into the P scarcity debate here if I were you
Response: Thank you so much for your comment. We are agree with your point, we have revised this line in the abstract see line 20. Yes biochar is the best alternative for the inorganic chemical fertilizers.
Comment:Introduction:
-You make a strong point for struvite and alternatives to mineral fertilizers. There is a recent study on heavy metals in mineral fertilizers (https://doi.org/10.1016/j.rser.2021.110740). Not using mineral phosphates may decrease heavy metal dissipation through fertilizers. This can also be added in the introduction to make your point even stronger
Response: Thanks for your suggestion. It’s done. Please see lines 58-63 of the revised manuscript.
Comment: Materials and Methods:
-Where exactly came the activated carbon from. A number of biowaste materials are presently considered to produce active carbon, so this could be important to some readers
-ok you have that in table 1 - good
Response: Thanks for your suggestion. The source of the carbon material is present in table 1 see line 197.
Comment: Results:
-line 235 check typo
-Fig 1 - I would call this extraction rate since you are not recovering it yet - it is with the biochar
-provide all figures in higher quality
Response: It’s done. We have corrected the typo mistake See line 237. Yes we may call it extraction rate but we have used recovery rate as most of the literature work has used word recovery rate instead of extraction rate. Hope you are satisfy with the point. We have revised the figures quality according your suggestion.
Comment: Discussion:
-include an assessment of the industrial applicability - these results are very interesting, now if I want to apply this on industrial scale, how would it look like?
Response: Thank you for valuable suggestion to make our manuscript discussion strong. We have added one separate paragraph in discussion on assessment of the industrial applicability of our study results. See lines 601-624.

Reviewer 2 Report
I find that this article is original and interesting. However, the English must be improved as it does not flow well.
Content
Page 1, lines 100-101: What do the authors mean by suitable price? “The purpose of a suitable price for the biogas slurry it is an essential aspect of the examination in order to attain and to promote the value of biogas slurry as a fertilizer”.
You would need to revise the principles of waste economics, and determinants of prices. I suggest authors read Lupton (2017). Moreover, the phrase is not clear (see form comments).
Form
The article should be very carefully proofread by a professional English-speaking editor, to avoid spelling & syntax errors. I have just given some examples in the first three pages to illustrate this. Without a clearer quality of the English language, this article is unpublishable.
Page 1 Abstract, line 33: a space is missing between “Cu2+, and CO32-.” And “While,…”.
Page 2 line 54, there seems to be an extra space in the text between “hexahydrate” and “ammonium”
Page 2, line 85, the comma is unnecessary, and the sentence should read as following: “While few researchers…”
Page 2, line 86, Approxmentaly should read as follows “approximately”.
Page 2, lines 90-91: I read “Several scholars [38-43] researchers”. I suggest authors either put “scholars” or “researchers” but not both.
Page 3, lines 95-97: the sentence is very vague, please rephrase. Would do the authors mean? I underlined the parts that render the sentence unclear. “Biogas slurry used as fertilizer is still focusing on the application effect of the biogas slurry are including enhancing crop productivity, endorsing product quality, improving crop resistance and enlightening the soil quality and fertility characteristics”.
Suggested reference
Lupton, S. (2017). Markets for waste and waste–derived fertilizers. An empirical survey. Journal of Rural Studies, 55, 83-99.
Author Response
Reviewer 2
Thank you for your valuable comments. We have revised this manuscript according to your suggestions. I hope that the revised manuscript would meet your requirements for Publication in Sustainability. Based on your suggestion and comments, we have carefully modified the manuscript (highlighted with yellow color for reviewer one and track changes for professional English editing). We wish that the revised manuscript will meet the standard for publication.
Comment: I find that this article is original and interesting. However, the English must be improved as it does not flow well.
Response: Thanks for comments and suggestion. The writing of manuscript has been improved, the language flow of the manuscript has been checked by the native professional speaker.
Comment: Content Page 1, lines 100-101: What do the authors mean by suitable price? “The purpose of a suitable price for the biogas slurry it is an essential aspect of the examination in order to attain and to promote the value of biogas slurry as a fertilizer”. You would need to revise the principles of waste economics, and determinants of prices. I suggest authors read Lupton (2017). Moreover, the phrase is not clear (see form comments).
Response: Thanks for suitable comment. Here suitable price denotes the value addition and worth of the biogas slurry. We have revised the sentence see line 105 of the revised manuscript.
Comment: The article should be very carefully proofread by a professional English-speaking editor, to avoid spelling & syntax errors. I have just given some examples in the first three pages to illustrate this. Without a clearer quality of the English language, this article is unpublishable.
Response: Thank you for valuable comments. We have carefully rechecked the whole manuscript to avoid unnecessary errors in the revised manuscript. The writing of manuscript has been improved, the language flow of the manuscript has been checked by the native speaker in this document, and most of the errors and inconsistencies has been improved.
Comment:Page 1 Abstract, line 33: a space is missing between “Cu2+, and CO32-.” And “While,…”.
Response: It’s done according to your suggestions. Please see line 33 in the abstract of the revised manuscript.
Comment: Page 2 line 54, there seems to be an extra space in the text between “hexahydrate” and “ammonium”
Response: It’s done according to your suggestions. Please see line 54 of the revised manuscript.
Comment: Page 2, line 85, the comma is unnecessary, and the sentence should read as following: “While few researchers…”
Response: It’s done according to your suggestions. Please see line 91 of the revised manuscript.
Comment: Page 2, line 86, Approxmentaly should read as follows “approximately”.
Response: It’s done according to your suggestions. Please see page 2 line 92 of the revised manuscript.
Comment: Page 2, lines 90-91: I read “Several scholars [38-43] researchers”. I suggest authors either put “scholars” or “researchers” but not both.
Response: It’s done according to your suggestions. Please see page 2 line 96 of the revised manuscript. We have deleted word researcher and used scholars.
Comment: Page 3, lines 95-97: the sentence is very vague, please rephrase. Would do the authors mean? I underlined the parts that render the sentence unclear. “Biogas slurry used as fertilizer is still focusing on the application effect of the biogas slurry are including enhancing crop productivity, endorsing product quality, improving crop resistance and enlightening the soil quality and fertility characteristics”.
Response. It’s done according to your suggestions. Please see page 3 line 101-103 of the revised manuscript. This sentence has been cleared by the professional native speaker.
Comment:Suggested reference
Lupton, S. (2017). Markets for waste and waste–derived fertilizers. An empirical survey. Journal of Rural Studies, 55, 83-99.
Response. It’s done according to your suggestions. We have also cited this reference in the revised manuscript.

Round 2
Reviewer 1 Report
Excellent - you addressed my comments - from my end this is fine now and I hope to see this being published soon. Great working with you.